# LIFT ME UP: THE IMPACT OF LIFTINGS ON HYPERGRAPH NEURAL NETWORKS

**Marco Montagna, Simone Scardapane & Lev Telyatnikov** *
Sapienza Università di Roma
Piazzale Aldo Moro 5, 00185 Roma, Italia
{marco.montagna, simone.scardapane, lev.telyatnikov}@uniroma1.it

## ABSTRACT

Hypergraph neural networks (HNNs) have become a powerful tool for modeling higher-order interactions in relational data. However, most HNN methods assume that the hypergraph structure is given. Whenever the data originates from a graph or a point cloud (which is common in practice) this requires a transformation step known as *lifting*. Despite its crucial role, the lifting process remains largely understudied and is often handled via ad hoc heuristics. In this work, we present the first systematic evaluation of hypergraph lifting strategies. We study seven diverse lifting methods and assess their impact on downstream classification tasks across a variety of datasets and three state-of-the-art hypergraph models. Moreover, we compare these lifting-based approaches against standard graph neural networks, demonstrating that finding the appropriate higher-order structure allows hypergraph models to outperform traditional graph baselines. Notably, our findings reveal that the choice of lifting often has a greater impact on performance than the choice of model architecture. While some liftings perform better than others, no single lifting consistently dominates on all datasets. These results suggest that further advances in hypergraph learning may come less from architectural innovations and more from better ways of constructing hypergraph structures.

## 1 INTRODUCTION

Despite their promise, hypergraph neural networks (HNNs) remain undercut by a simple, underexplored question: *where do hypergraphs come from?* This work presents the first systematic evaluation of hypergraph liftings, the procedures that transform graphs or point clouds into hypergraphs, and reveals that this step often plays a more decisive role in downstream performance than the model architecture itself. Across seven lifting strategies, three state-of-the-art hypergraph models, and a diverse suite of datasets, our results consistently show that the quality of the hypergraph construction strongly determines the final performance. Furthermore, we benchmark these lifting-based hypergraph models against standard Graph Neural Networks (GNNs). Our results demonstrate that finding the right higher-order structure allows HNNs to outperform traditional graph baselines, showing that the provided graph topology is not always the optimal structure for learning. These findings call into question the current emphasis on architectural novelty and suggest that progress in hypergraph learning may depend more on structure design than previously recognized.

Hypergraphs are powerful tools for modeling higher-order interactions, allowing one to move beyond pairwise edges to capture group-wise dependencies. They have been successfully applied in a wide range of domains, including recommender systems Xia et al. (2022), bioinformatics Sun et al. (2008), and knowledge graphs Fatemi et al. (2023). As a result, there has been a surge of interest in developing expressive and scalable hypergraph neural networks (HNNs) Zhou et al. (2006); Feng et al. (2019); Chien et al. (2022). Yet, a fundamental assumption underlies much of this work: that the hypergraph structure is provided. In practice, however, data typically arrive as pairwise graphs, point clouds, or even raw feature matrices. To apply HNNs, one must first transform this input into a hypergraph, a process known as *lifting*. While many lifting strategies exist (e.g., k-hop neighborhoods, knn algorithms) Dai & Gao (2023), their influence on downstream performance has received

---

*Work done while enrolled as a PhD in Sapienza.

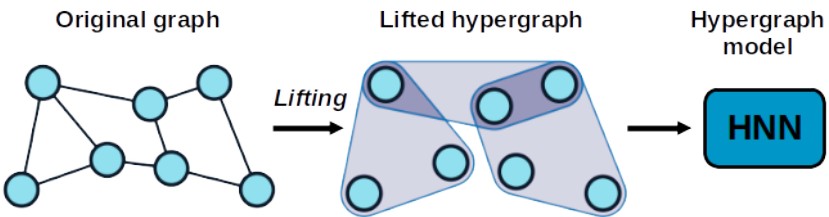

Figure 1: **Overview of the lifting pipeline.** A graph or point cloud is transformed into a hypergraph via a lifting strategy $\phi$, which is then processed by a Hypergraph Neural Network.

limited systematic attention. In contrast, we approach liftings not as arbitrary preprocessing choices but as *critical design decisions* that deserve careful evaluation. Rather than using artificial proxy tasks, such as hyperedge prediction, we directly measure the impact of each lifting on real downstream performance across both node and graph classification benchmarks. Surprisingly, we find that certain liftings (e.g., EK-Hop) generalize remarkably well across models and domains, while others, despite theoretical motivations, underperform in practice.

## 2  RELATED WORKS

Hypergraphs have emerged as a powerful abstraction for modeling higher-order relationships in complex systems, enabling generalizations of message passing and learning beyond pairwise interactions. A substantial body of work has focused on developing expressive hypergraph neural networks (HGNNs) Zhou et al. (2006); Feng et al. (2019), leveraging a variety of mechanisms, from incidence-based convolutions Huang & Yang (2021) to attention-based models Chien et al. (2022). These approaches typically assume that the hypergraph structure is provided, and relatively little attention has been paid to the upstream construction of the hypergraph itself.

The task of constructing hypergraphs from raw graph data, which we refer to as *hypergraph lifting*, is often overlooked or addressed through heuristics. Prior approaches include simple structural transformations such as $k$-hop or nearest-neighbor-based grouping Dai & Gao (2023). While these methods can be effective, they are often chosen arbitrarily and lack systematic evaluation. Hypergraph liftings can be alternatively viewed through the lens of hyperedge prediction, where the objective is not to optimize performance on a downstream task, but to explicitly infer the hypergraph structure itself, typically by predicting which sets of nodes should form hyperedges. Several works have explored this direction Li et al. (2013); Monti et al. (2017). Coordinated Matrix Minimization Zhang et al. (2018b) uses nonnegative matrix factorization and least-squares matching over the vertex adjacency space to recover missing hyperedges. Neural Hypergraph Prediction (NHP) Yadati et al. (2020) employs a GCN-based encoder Kipf & Welling (2017) to score candidate node sets, learning to predict hyperedges directly. These methods treat hyperedge construction as a standalone learning problem, decoupled from any downstream task, and often require supervision on the hyperedges.

A distinct line of research focuses on *Hypergraph Structure Learning (HSL)*, where the goal is to optimize the hypergraph topology jointly with the model parameters to improve downstream task performance. Methods such as Dynamic Hypergraph Structure Learning (DHSL) Zhang et al. (2018a) and Hypergraph Structure Learning (HGSL) Cai et al. (2022) propose mechanisms to dynamically generate or refine hyperedges during training. However, these approaches typically assume an initial hypergraph structure is provided as input, which they subsequently refine, re-weight, or expand. Therefore, in our setting, where the input data consists of graphs or point clouds, these methods are not alternatives to lifting but rather downstream optimizations that must be applied *after* an initial lifting step has transformed the raw data into a hypergraph.

While existing studies have compared different hypergraph models Telyatnikov et al. (2025a), they typically fix the hypergraph structure, either by using datasets already structured as hypergraphs or by adopting a single lifting strategy, and focus on evaluating model performance. A recent ICML challenge Bernárdez et al. (2024) invited submissions of lifting methods but did not assess their

Table 1: Summary of the various liftings considered in this work. The number of hyperedges produced by the `FR` lifting depends on the number of connected components that are found after removing the edges with lower curvature.

| Name | Connectivity | Features | # Hyperedges |
|---|---|---|---|
| `KNN` | No | Yes | $|V|$ |
| `K-Hop` | Yes | No | $|V|$ |
| `EK-Hop` | Yes | No | $K|V|$ |
| `FR` | Yes | No | Varies |
| `Kernel` | Yes | Yes | $|V|$ |
| `Mapper` | Yes | Yes | $|\mathcal{U}|$ |
| `MM` | Yes | Yes | $|V|$ |

impact on downstream tasks. To the best of our knowledge, this is the first work to systematically evaluate a broad range of hypergraph lifting strategies in terms of their downstream performance.

## 3 BACKGROUND

A graph $\mathcal{G}$ is composed of nodes and edges and can be written as $\mathcal{G} = (V, E)$, where $V$ is the set of nodes and $E$ the set of edges, each connecting a pair of nodes. One common way to represent the structure of a graph is through its adjacency matrix $A \in \{0,1\}^{|V| \times |V|}$, where $A_{ij} = 1$ if there is an edge between nodes $v_i$ and $v_j$, and $0$ otherwise.

**Hypergraphs** A hypergraph is a generalization of a graph in which edges, called *hyperedges*, can connect any number of nodes, not just two.

**Definition 1** *A hypergraph $\mathcal{H}$ is a tuple $\mathcal{H} = (V, \mathcal{E}_H)$, where $\mathcal{E}_H$ is a non-empty subset of the power set $\mathcal{P}(V) \setminus \{\emptyset\}$.*

The *cardinality* of a hyperedge $h \in \mathcal{E}_H$ is the number of nodes it contains. Under this definition, standard graphs can be viewed as a special case of hypergraphs in which all hyperedges have cardinality two (or one in the case of self-loops). In many applications, it is useful to associate features to both nodes and hyperedges. By defining $d_v$ and $d_h$ as the dimension of the node and hyperedge features respectively, we denote the node feature matrix as $X_V \in \mathbb{R}^{|V| \times d_v}$ where each row corresponds to the features of a node, and the hyperedge feature matrix as $X_H \in \mathbb{R}^{|\mathcal{E}_H| \times d_h}$.

**Liftings** A *lifting* refers to a function that maps graphs (or point clouds) to hypergraphs. Following the existing literature Bernárdez et al. (2024), we distinguish between different categories of liftings:

*Feature-based liftings* use node features to define hyperedges. These are especially useful when the input is a point cloud or unstructured data with no predefined connectivity. Despite not leveraging edge information, they are applicable in scenarios where node features encode rich relations.

*Connectivity-based liftings* rely solely on the graph structure to define hyperedges. These are appropriate when connectivity plays a central role in the task or when node features are sparse or uninformative, for instance, when features are simple one-hot encodings or absent, requiring the model to learn purely from structural embeddings Grover & Leskovec (2016).

*Hybrid liftings* incorporate both node features and connectivity. These methods aim to combine the structural context of the graph topology with the information contained in the node features.

## 4 HYPERGRAPH LIFTINGS

This section introduces the seven hypergraph liftings evaluated in our study. These methods span a variety of ideas: some rely purely on graph connectivity, others use node features, and several combine both. While a subset of these liftings are commonly used in the literature (e.g., `KNN` and `K-Hop` Dai & Gao (2023)), we introduce the `Exclusive K-Hop (EK-Hop)` specifically

for this work, while `FR`, `Kernel`, `Mapper`, and `MM` liftings were proposed at the ICML 2024 Topological Deep Learning Challenge Bernárdez et al. (2024) and have been made available in the TopoBench library Telyatnikov et al. (2025a). Table 1 lists the liftings and their basic properties.

**KNN Lifting**  The *K-Nearest Neighbors* lifting is a feature-based method that constructs hyperedges by grouping nodes with similar features. For each node $v \in V$, a hyperedge is formed by identifying the $K$ nodes whose feature vectors are closest to $v$ in Euclidean space. The node $v$ itself is included in its own hyperedge. Formally, the hyperedge associated with node $v$ is defined as:

$$h_v = \{v_j \in V \mid v_j \text{ is among the } K \text{ nearest neighbors of } v\}.$$

This method is commonly used in practice, particularly in homophilic datasets, where nodes with similar features are expected to share connections Zhu et al. (2020).

**K-Hop Lifting**  The *K-Hop* lifting is a connectivity-based method that constructs hyperedges using only the graph structure. To define it formally, let $d_{\mathcal{G}}(u, v)$ denote the shortest path distance (number of edges) between two nodes $u$ and $v$ in the graph $\mathcal{G}$. For each node $v_i \in V$, the lifting constructs a single hyperedge containing all nodes reachable within $K$ steps. Using the distance notation, this can be written compactly as:

$$h_{v_i} = \{v_j \in V \mid d_{\mathcal{G}}(v_i, v_j) \leq K\}.$$

This lifting is inspired by $K$-layer message-passing neural networks, where the receptive field of a node expands to include its $K$-hop neighborhood. By explicitly grouping these nodes into a single hyperedge, the lifting captures the aggregate local information available to the node.

**EK-Hop Lifting**  This lifting is novel and has been developed for this work. The *Exclusive K-Hop* lifting modifies the standard `K-Hop` approach by avoiding the aggregation of neighbors at different distances. Instead, for each node $v_i \in V$, it creates $K$ distinct hyperedges, one for each specific hop distance $k \in \{1, \ldots, K\}$. Formally, the hyperedge corresponding to the $k$-th hop of node $v_i$ is defined as:

$$h_{v_i, k} = \{v_j \in V \mid d_{\mathcal{G}}(v_i, v_j) = k\} \cup \{v_i\}.$$

Unlike `K-Hop`, which merges the 1-hop, 2-hop, ..., and $K$-hop neighbors into one set, `EK-Hop` stratifies the neighborhood. Each hyperedge $h_{v_i, k}$ connects the central node $v_i$ exclusively to nodes that are exactly $k$ steps away. This separation allows the downstream model to potentially learn different weights for immediate neighbors versus distant ones, preventing the "over-squashing" of local and global information into a single channel.

**FR Lifting**  The *Forman-Ricci Curvature* lifting is a connectivity-based method that leverages discrete Forman-Ricci (FR) curvature to extract cohesive regions of a graph. The FR curvature of an edge provides a notion of how structurally well-integrated it is within its local neighborhood Weber et al. (2018). Intuitively, edges with high positive curvature tend to lie within dense clusters, whereas edges with negative curvature often bridge sparsely connected parts of the graph.

This lifting proceeds in three steps: (1) *Curvature computation:* Compute the FR curvature for each edge $e_{ij} \in E$ using the definition in Weber et al. (2018). (2) *Edge pruning:* Discard all edges whose curvature falls below or above a specified threshold, typically chosen to preserve a fixed percentage of the highest-curvature edges. (3) *Hyperedge formation:* Identify the connected components of the resulting pruned graph. Each connected component is then treated as a hyperedge. By retaining only the edges with high curvature, this lifting aims to isolate tightly connected clusters, forming hyperedges that reflect coherent communities within the graph. Meanwhile, by retaining edges with low curvature, this lifting can isolate the edges that connect clustered regions.

**Kernel Lifting**  The *Kernel Lifting* is a hybrid approach that incorporates both connectivity and feature information to define a distance between nodes. It relies on a kernel function of the form

$$\kappa(v_i, v_j, \mathcal{G}) = C\big(\kappa_v(v_i, v_j, \mathcal{G}), \kappa_x(v_i, v_j)\big),$$

where: (1) $\kappa_v$ is a *connectivity kernel* that uses the graph structure to return the distance between two nodes. The graph kernels considered in this work are: the identity $\kappa_v = 1$, the heat kernel $\kappa_v = \exp(-t\mathcal{L}_{ij})$ based on the heat transfer equation, and the Matérn kernel $\kappa_v = \left(\frac{2\nu}{\kappa^2 \mathcal{I}} + \mathcal{L}\right)^{-\nu}$ based

on the Matérn covariance function. (2) $\kappa_x$ is a *feature kernel* that calculates the similarity between nodes using their features. To calculate similarities between nodes, three options are explored: the cosine and Euclidean similarities, and the identity, in which all pairs are scored as 1. (3) $C$ is a combination function; in this work, the sum that merges the two similarity measures into one kernel.

Once $\kappa(v_i, v_j, \mathcal{G})$ is computed, it induces a similarity score between every pair of nodes. The lifting then proceeds similarly to the `KNN` lifting: for each node $v_i$, a hyperedge is formed by selecting the $K$ nodes with the highest similarity to $v_i$.

This lifting generalizes `KNN` by allowing for richer, more flexible similarity measures. By tuning the kernel components, it is possible to emphasize connectivity, features, or both. However, its performance heavily depends on the choice of kernel and parameters, which may need to be tailored to the specific tasks or datasets.

**Mapper Lifting**    The *Mapper Lifting* is inspired by the Mapper algorithm from Topological Data Analysis Singh et al. (2007). It constructs hyperedges by applying a scalar-valued *filter function* $g : V \rightarrow [a, b]$ to the nodes, which maps each node to a value within a real interval. This lifting can use the graph connectivity or the node features based on the filter function.

A cover $\mathcal{U}$ of the interval $[a, b]$ is then defined, consisting of $N$ overlapping or non-overlapping subintervals. The amount of overlap between intervals is a tunable parameter. Each interval $U \in \mathcal{U}$ defines a preimage set of nodes $h_U = \{v \in V \mid g(v) \in U\}$, which forms a hyperedge. In other words, all nodes whose filter values fall within the same interval are grouped into a hyperedge. To ensure sufficient connectivity and expressiveness, especially when $N$ is small and hyperedges may be coarse, the original edges are also added as hyperedges.

The choice of filter function $g$ strongly influences the structure of the resulting hypergraph. It can be based on node features, spectral embeddings, centrality scores, or other domain-specific priors. In this work, we tested: (1) the first Laplacian Eigenvector Positional Encoding Dwivedi et al. (2023) as the filter function $g$, i.e., the component $k = 1$ of the spectral embedding of the graph, (2) the first singular vector from a Singular Value Decomposition (SVD) of the node feature matrix Eckart & Young (1936), (3) the first principal component from a Principal Component Analysis (PCA) Jolliffe (2002), and (4) the sum of the node features.

**MM Lifting**    The *Modularity Maximization (MM) Lifting* constructs hyperedges by first identifying communities in the graph using spectral clustering on the modularity matrix Newman (2006). The modularity matrix $B$ is defined as $B_{ij} = A_{ij} - d_i d_j / (2|E|)$, where $|E|$ is the total number of edges in the graph, and $d_i$ and $d_j$ are the degrees of nodes $v_i$ and $v_j$ respectively. The matrix $B$ captures the deviation of edge density from what is expected under a random graph with the same degree distribution. To detect communities, spectral clustering is performed on $B$: the top eigenvectors are computed and, once stacked, are used as features for the nodes of the graph. Then, K-means clustering is used to partition the graph into a predefined number of communities. Once the communities are identified, for each node $v_i$, a hyperedge is created by selecting its $K$ nearest neighbors within the same community $\mathcal{C}_{v_i}$ based on a predefined distance metric—specifically, the Euclidean distance is considered in this work $h_{v_i} = \{v_j \in \mathcal{C}_{v_i} \mid v_j \text{ is among the } K\text{-NNs of } v_i\}$. This lifting combines structural community detection with feature-based similarity, aiming to group nodes that are both locally similar and globally coherent within the graph's community structure. We propose to add the original graph edges as 2-hyperedges to help the resulting hypergraph be connected. We treated this addition as a hyperparameter choice and tested its contribution.

## 5    EXPERIMENTAL SETUP

**Hypergraph Models**    We have selected three state-of-the-art hypergraph neural networks that cover a variety of architectural design principles. These models have been shown to perform well across different hypergraph learning tasks and serve as strong baselines in recent literature Chien et al. (2022); Huang & Yang (2021); Wang et al. (2023); Telyatnikov et al. (2025a). By including models that span diverse architectural paradigms, such as attention-based aggregation (`AllSetTransformer`), spectral and residual-based message passing (`UniGCNII`), and symmetry-preserving aggregation schemes (`ED-HNN`), our experimental setup enables a broad evaluation of hypergraph liftings. These models capture different inductive biases and structural assump-

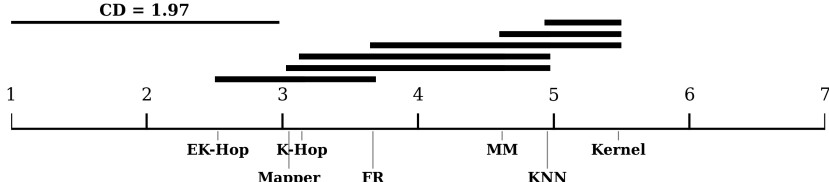

Figure 2: **Critical Difference (CD) diagram of lifting performance.** The diagram plots the average rank of each lifting strategy across all datasets and models, where lower ranks (further left) indicate better performance. Horizontal bars connect groups of liftings that are not statistically significantly different from each other according to a Nemenyi post-hoc test at a significance level of $\alpha = 0.05$.

tions, making them complementary choices for assessing the robustness of the lifting strategies. Appendix C offers an overview of the different models.

**Datasets**   To evaluate the effectiveness and generalizability of the proposed liftings, we curated a set of benchmark datasets that span different domains, structural properties, and learning tasks. Our goal was to test the liftings across a broad spectrum of settings, with a particular focus on varying levels of homophily and underlying graph structures. Homophily is a measure of the tendency of nodes with similar labels to be connected, and it has been shown to affect the performance of hypergraph neural networks significantly Telyatnikov et al. (2025b). To account for this, our dataset selection includes both homophilic and heterophilic graphs (graphs with high and low levels of homophily). We also consider datasets composed of a single large graph and ones consisting of multiple smaller graphs. These structural differences are particularly relevant when designing hypergraph liftings, since a lifting operates at the graph level and its effect can vary significantly depending on the initial structure. The selected datasets include: (1) *Cora*, *Citeseer*, and *Pubmed* McCallum et al. (2000); Giles et al. (1998); Sen et al. (2008): These are canonical citation network benchmarks characterized by high homophily. The task is node classification, and each dataset contains a single connected graph. (2) *Amazon* and *Empire* Platonov et al. (2022): These represent node classification benchmarks in a heterophilic regime, where nodes are often connected to other nodes with different labels. These datasets test the ability of hypergraph models and liftings to generalize beyond the homophilic assumption. (3) *Mutag* Debnath et al. (1991) and *Proteins* Borgwardt et al. (2005): These are graph classification benchmarks composed of multiple small, disconnected graphs. Each graph represents a molecular structure (Mutag) or a protein (Proteins), and the classification task is to assign a label to the entire graph. By covering a range of structural patterns and learning scenarios, this suite of datasets enables a comprehensive and stress-tested evaluation of the proposed liftings and their compatibility with different models.

**Training and Evaluation Protocol**   We adopt the TopoBench Telyatnikov et al. (2025a) evaluation protocol, assessing performance via accuracy across all node and graph classification tasks to maintain consistency with standard benchmarks. To ensure fair comparisons, we employ a standardized training pipeline for all model-lifting-dataset combinations. Each dataset is evaluated using five random splits with a 50%/25%/25% train/validation/test ratio. To compare our results against standard graph models, we utilize the baseline results provided by the TopoBench framework. Since our experiments use the same library and data loading pipeline the results reported for the graph baselines are directly comparable to our hypergraph lifting results.

Model hyperparameters were selected via a grid search. For general architectural parameters, we explored hidden dimensions $h \in 32, 64, 128$ and learning rates $\eta \in 10^{-2}, 10^{-3}$, applying an initial linear projection to map input features to the hidden dimension. In addition to these standard parameters, the topological liftings introduce specific hyperparameters governing their behavior (e.g., kernel parameters). Due to the diversity of these lifting-specific parameters, their respective search spaces are detailed in Appendix E. The optimal configuration for each experiment was selected based on the highest average validation accuracy across five random seeds. The results reported in the following sections correspond to the mean accuracy and standard deviation computed on the test sets using these best-performing configurations.

Table 2: Performance comparison across models and liftings. OOT means that the lifting process exceeded the maximum allowed time of 180 minutes.

| Model | Method | Cora | Citeseer | Pubmed | Empire | Proteins | Mutag | Amazon |
|-------|--------|------|----------|--------|--------|----------|-------|--------|
| GCN | | **87.09 ± 0.20** | **75.53 ± 1.27** | 89.40 ± 0.30 | 78.16 ± 0.32 | 75.70 ± 2.14 | 69.79 ± 6.80 | 49.56 ± 0.55 |
| GIN | | 87.21 ± 1.89 | 73.73 ± 1.23 | 89.29 ± 0.41 | 79.56 ± 0.20 | 75.20 ± 3.30 | **79.57 ± 6.13** | 49.16 ± 1.02 |
| GAT | | 86.71 ± 0.95 | 74.41 ± 1.75 | **89.44 ± 0.24** | **84.02 ± 0.51** | **76.34 ± 1.66** | 72.77 ± 2.77 | **50.17 ± 0.59** |
| AST | KNN | 75.16 ± 1.08 | 68.76 ± 1.46 | 87.35 ± 0.57 | 65.83 ± 0.41 | **76.70 ± 1.64** | 72.77 ± 5.93 | 45.41 ± 1.11 |
| | K-Hop | 87.36 ± 0.81 | 74.24 ± 1.10 | 89.38 ± 0.30 | 77.04 ± 0.85 | 76.34 ± 2.13 | 72.34 ± 7.25 | 48.29 ± 0.75 |
| | EK-Hop | **87.59 ± 0.80** | **76.30 ± 4.84** | **89.45 ± 0.29** | 77.09 ± 0.84 | 76.34 ± 2.13 | 71.91 ± 7.30 | 48.38 ± 0.50 |
| | FR | 80.71 ± 1.15 | 72.10 ± 0.71 | 87.56 ± 0.44 | 69.83 ± 0.27 | 76.34 ± 1.99 | **82.13 ± 3.71** | 45.55 ± 0.58 |
| | Kernel | 86.85 ± 0.72 | 72.41 ± 0.89 | OOT | OOT | 75.13 ± 1.96 | 72.77 ± 6.28 | OOT |
| | Mapper | 86.88 ± 1.12 | 74.84 ± 1.33 | 89.43 ± 0.56 | **80.66 ± 0.39** | 76.63 ± 1.58 | 72.77 ± 5.78 | **48.46 ± 0.65** |
| | MM | 78.67 ± 1.29 | 69.32 ± 1.10 | 87.60 ± 0.40 | 71.46 ± 0.30 | 76.56 ± 2.30 | 73.19 ± 6.68 | 45.65 ± 0.55 |
| UniGCNII | KNN | 69.01 ± 1.48 | 29.60 ± 0.90 | 87.44 ± 0.64 | 65.47 ± 0.40 | 76.13 ± 1.91 | 81.28 ± 2.49 | 46.88 ± 0.74 |
| | K-Hop | **86.12 ± 0.76** | **73.45 ± 1.40** | 88.50 ± 0.73 | **74.45 ± 0.58** | 76.49 ± 1.80 | 70.21 ± 7.00 | 46.09 ± 0.48 |
| | EK-Hop | **86.12 ± 0.76** | 73.42 ± 1.42 | **88.62 ± 0.43** | 74.29 ± 0.00 | 76.49 ± 1.80 | 82.55 ± 5.29 | 46.04 ± 0.00 |
| | FR | 78.43 ± 1.34 | 70.76 ± 1.82 | 87.34 ± 0.27 | 68.07 ± 0.22 | **76.56 ± 2.32** | **84.26 ± 6.68** | **47.48 ± 1.29** |
| | Kernel | 83.25 ± 1.48 | 69.34 ± 1.13 | OOT | OOT | 75.63 ± 1.54 | 80.85 ± 3.98 | OOT |
| | Mapper | 82.48 ± 1.45 | 73.28 ± 1.39 | 43.92 ± 8.86 | 65.92 ± 0.25 | 76.34 ± 1.66 | 81.28 ± 5.11 | 45.31 ± 0.67 |
| | MM | 70.84 ± 1.03 | 35.87 ± 1.63 | 87.43 ± 0.45 | 65.39 ± 0.39 | 76.06 ± 1.97 | 82.13 ± 4.39 | 46.87 ± 0.85 |
| ED-HNN | KNN | 72.53 ± 1.37 | 69.03 ± 1.01 | 87.65 ± 0.42 | 65.66 ± 0.40 | **76.49 ± 1.43** | 80.85 ± 2.70 | 44.09 ± 0.76 |
| | K-Hop | 86.65 ± 0.73 | 73.95 ± 1.26 | 89.41 ± 0.29 | 72.81 ± 0.50 | 75.20 ± 1.25 | 71.49 ± 5.49 | 46.75 ± 0.87 |
| | EK-Hop | **86.88 ± 1.13** | 73.95 ± 1.26 | **89.52 ± 0.39** | 72.84 ± 0.53 | 75.34 ± 0.89 | 71.06 ± 7.20 | **46.76 ± 0.91** |
| | FR Curv | 76.43 ± 0.44 | 70.76 ± 0.86 | 87.53 ± 0.37 | 68.74 ± 0.52 | 76.20 ± 2.46 | **82.55 ± 2.09** | 45.61 ± 0.76 |
| | Kernel | 85.11 ± 1.28 | 70.83 ± 1.37 | OOT | OOT | 75.48 ± 2.39 | **82.55 ± 3.81** | OOT |
| | Mapper | 84.84 ± 0.79 | **74.02 ± 1.01** | 88.05 ± 0.81 | **74.01 ± 0.45** | 76.20 ± 2.07 | 77.87 ± 4.78 | 46.07 ± 0.64 |
| | MM | 74.62 ± 1.26 | 68.84 ± 1.51 | 87.75 ± 0.48 | 67.60 ± 0.69 | 75.77 ± 1.91 | 82.13 ± 2.17 | 44.48 ± 0.68 |

The experiments were conducted on a server running `Ubuntu 24.04`, equipped with an `Intel i9-13900K` CPU and an `NVIDIA GeForce RTX 4090` GPU (24GB VRAM). The software environment included CUDA 12.4, PyTorch 2.3.0, and PyTorch Lightning 2.4.0. The liftings were interrupted if they took more than 180 minutes on this machine. All code and reproduction instructions will be made publicly available upon acceptance.

## 6 RESULTS

To compare the relative performance of the different liftings across multiple datasets and models, we employ a Critical Difference (CD) diagram Demšar (2006), shown in Figure 2. This visualization is based on the Friedman test followed by a Nemenyi post-hoc analysis. The diagram plots the average rank of each lifting method on the x-axis (lower is better). Groups of liftings that are not statistically significantly different from one another (at $p < 0.05$) are connected by a bold horizontal bar. We chose this ranking-based evaluation because it provides a robust way to aggregate results across heterogeneous datasets where baseline accuracy varies significantly, making it less sensitive to outliers than simple average accuracy. Detailed numerical ranking tables for each model are provided in Appendix D.

The CD diagram reveals that `EK-Hop` achieves the best (lowest) average rank, followed closely by `Mapper` and `K-Hop`. However, the horizontal bar connecting `EK-Hop`, `Mapper`, `K-Hop`, and `FR` indicates that the performance differences between these top methods are not statistically significant at the 95% confidence level. This aligns with our statistical analysis, confirming that while `EK-Hop` is a strong default, it does not strictly dominate the other top-performing strategies in a statistically consistent manner. It is worth noting that the high standard deviation observed on graph classification tasks (e.g., Mutag, Proteins) contributes significantly to this statistical overlap, making it difficult to establish dominance on these smaller datasets despite clear trends in average rank. Full performance metrics, including accuracy and standard deviation across runs, are provided in Table 2. Several consistent patterns emerge from these results.

**Finding 1** Lifting-based HNNs outperform standard GNNs by escaping the limitations of the raw graph topology. Our comparison reveals that hypergraph models, when paired with appropriate liftings, consistently surpass standard Graph Neural Network baselines (GCN, GAT, GIN) across the majority of datasets. This indicates that the raw input graph is often a suboptimal structure. By treating the input topology not as a fixed ground truth but as a starting point for structural optimization, lifting allows HNNs to capture dependencies that standard graph models miss.

**Finding 2** Model selection yields marginal gains compared to lifting selection. In other words, the lifting strategy often has a larger influence on downstream performance than the specific model architecture. This highlights a potentially underappreciated aspect of hypergraph learning: while much of the recent work in topological deep learning has focused on architectural innovation, our results suggest that substantial gains may instead be achieved by improving the way graphs are lifted into higher-order representations.

**Finding 3** The `EK-Hop` lifting demonstrates strong and robust performance across a diverse range of scenarios. It performs well across both homophilic and heterophilic datasets, and proves effective regardless of the task being node or graph classification. We hypothesize that this robustness stems from the *stratification* of the neighborhood: unlike `K-Hop`, which aggregates all neighbors into a single "bucket" (often leading to over-squashing of information), `EK-Hop` creates distinct hyperedges for each distance $k$. This allows the model to learn different importance weights for immediate versus distant neighbors, effectively disentangling local signal from global context. However, `EK-Hop` is not universally optimal: for instance, it ranks lowest on Mutag when used with `AST` and `ED-HNN`, highlighting that no lifting method consistently dominates across all settings.

**Finding 4** `AllSetTransformer` often emerges as the best-performing model when, for each dataset, we select the lifting that yields the highest accuracy. In this per-dataset best-case analysis, `AST` mostly outperforms the other models. However, this advantage disappears when comparing models under the same fixed lifting: in those direct comparisons, `AST` is not always superior. This discrepancy suggests that evaluating hypergraph models using a single lifting can lead to misleading conclusions. In contrast, comparing models *across* liftings provides a more robust assessment.

## 6.1 COMPLEXITY ANALYSIS

Although a lifting can be implemented as a one-time preprocessing step, the computational cost can be prohibitive for practical applications. We report the empirical preprocessing times in Table 3. The results highlight a clear scalability gap: while `K-Hop` and `EK-Hop` scale efficiently to larger graphs, the `Kernel` and `MM` liftings failed to complete within the 180-minute time limit on the larger datasets (*Pubmed*, *Empire*, *Amazon*). This effectively renders these methods unusable for large-scale graph learning without significant algorithmic optimization or approximation.

In terms of space complexity, the trade-offs are more nuanced. In standard graph learning, memory usage scales with the number of edges $|E|$. For liftings like `K-Hop` and `EK-Hop`, the number of hyperedges is linear with respect to nodes ($|\mathcal{E}_H| \propto |V|$). However, the critical factor for VRAM usage is not just the number of hyperedges, but the *incidence volume*, the total number of node-hyperedge connections. For dense graphs or large $K$, the hyperedges become highly populated, causing the incidence matrix density to approach $O(K|V|)$. Consequently, while the *structure* is compact in terms of edge counts, the memory required to store the incidences can still grow significantly.

## 7 CONCLUSION

The prevailing narrative in hypergraph learning has largely focused on designing more expressive neural architectures. Our findings challenge this paradigm, demonstrating that *how* we construct the hypergraph often matters more than which model we use to process it. By systematically benchmarking diverse lifting strategies, we reveal that the performance gap between different liftings frequently eclipses the variance between state-of-the-art architectures. Furthermore, our comparison with standard GNNs provides compelling evidence that the raw graph topology is rarely the optimal substrate for message passing. Lifting graphs into hypergraphs acts as a powerful form of structural optimization, allowing HNNs to capture long-range dependencies and mitigate topological bottlenecks that limit pairwise graph baselines.

While no single strategy serves as a universal solution, our proposed `EK-Hop` lifting emerges as a robust default, offering a strong balance of performance and efficiency across both homophilic and heterophilic regimes. These results point toward a necessary shift in research focus: moving from architectural engineering to *data-centric* structure engineering. Future work should move beyond ad hoc heuristics to establish a rigorous understanding of how specific liftings alter the spectral properties of the graph, offering principled guidelines on matching liftings to dataset characteristics.

Finally, this work sets the stage for the next frontier: *end-to-end structure learning*. While static liftings offer scalability, the substantial gains observed here suggest that the optimal structure lies in the latent space of the data itself. Although differentiable structure learning remains computationally expensive due to the combinatorial explosion of possible hyperedges Cai et al. (2022); Zhang et al. (2018a), our results validate the premise of this direction. If simple, static rewiring can unlock state-of-the-art performance, developing scalable, differentiable liftings that adapt the topology during training is likely the key to unlocking the full potential of geometric deep learning.

ACKNOWLEDGMENTS

This work was partially supported by the Sapienza University of Rome through the research grant RG123188B3EF6A80 (CENTS). We acknowledge ISCRA for awarding this project access to the LEONARDO supercomputer, owned by EuroHPC Joint Undertaking, hosted by CINECA (Italy).

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

## A  Lifting algorithms

This section provides the full pseudocode for each of the hypergraph lifting methods evaluated in the main paper. These algorithms transform input graphs into hypergraphs using a variety of strategies based on connectivity, node features, or both. The pseudocode is presented in a standardized format to facilitate clarity, reproducibility, and ease of implementation.

---

**Algorithm 1** `EK-Hop` Lifting

---

**Require:** Adjacency matrix $A \in \{0,1\}^{n \times n}$, hop radius $K$
**Returns:** Hyperedge set $\mathcal{E}_H$
 1: $R_0 \leftarrow I$
 2: **for** $k \leftarrow 1$ to $K$ **do**
 3:   $R_k \leftarrow A^k$                ▷ Nodes reachable in exactly $k$ hops
 4:   $M_k \leftarrow R_k > 0$              ▷ Binary mask of $k$-hop neighbors
 5:   **for** $l \leftarrow 1$ to $k-1$ **do**          ▷ Exclude nodes already seen
 6:     $M_k \leftarrow M_k \wedge \neg M_l$
 7:   **end for**
 8: **end for**
 9: **for** $i \leftarrow 1$ to $n$ **do**
10:   **for** $k \leftarrow 1$ to $K$ **do**
11:     $h_{i,k} \leftarrow \{v_j \in V \mid (M_k)_{ij}\} \cup \{v_i\}$
12:     $\mathcal{E}_H \leftarrow \mathcal{E}_H \cup \{h_{i,k}\}$
13:   **end for**
14: **end for**
15: **return** $\mathcal{E}_H$

---

**Algorithm 2** `Forman-Ricci Curvature` Lifting

---

**Require:** Graph $G = (V, E)$, curvature threshold $\tau \in \mathbb{R}$
**Returns:** Hyperedge set $\mathcal{E}_H$
 1: **for** each edge $e_{ij} \in E$ **do**
 2:   Compute Forman-Ricci curvature $\mathcal{F}(e_{ij})$
 3: **end for**
 4: $E' \leftarrow \{e_{ij} \in E \mid \mathcal{F}(e_{ij}) \geq \tau\}$         ▷ Prune low-curvature edges
 5: $G' = (V, E')$                  ▷ Construct pruned graph
 6: Compute connected components $\{C_1, \ldots, C_m\}$ of $G'$
 7: **for** each component $C_i$ **do**
 8:   $h_i \leftarrow C_i$
 9:   $\mathcal{E}_H \leftarrow \mathcal{E}_H \cup \{h_i\}$
10: **end for**
11: **return** $\mathcal{E}_H$

---

**Algorithm 3** `Kernel` Lifting

---

**Require:** Graph $\mathcal{G} = (V, E)$ with Laplacian $\mathcal{L}$, kernel temperature $t > 0$, neighborhood size $K$
**Returns:** Hyperedge set $\mathcal{E}_H$
 1: Connectivity kernel: $\kappa_v(v_i, v_j) = \exp(-t\mathcal{L}_{ij})$
 2: Feature kernel: $\kappa_x(v_i, v_j) = 1$
 3: Composite kernel: $\kappa(v_i, v_j) = \kappa_v(v_i, v_j) \cdot \kappa_x(v_i, v_j)$
 4: **for** each node $v_i \in V$ **do**
 5:   Compute similarity scores $\kappa(v_i, v_j)$ for all $v_j \in V$
 6:   Let $N_K(v_i) \leftarrow$ top $K$ nodes with highest $\kappa(v_i, v_j)$
 7:   $h_{v_i} \leftarrow N_K(v_i) \cup \{v_i\}$
 8:   $\mathcal{E}_H \leftarrow \mathcal{E}_H \cup \{h_{v_i}\}$
 9: **end for**
10: **return** $\mathcal{E}_H$

---

---

**Algorithm 4** `KNN` Lifting

---

**Require:** Feature matrix $X \in \mathbb{R}^{n \times d}$, # of neighbors $K$
**Returns:** Hyperedge set $\mathcal{E}_H$
1: **for** $i \leftarrow 1$ to $n$ **do**
2: $\quad S \leftarrow \text{argsort}_{j \neq i} \|X_i - X_j\|_2$
3: $\quad \mathcal{N}_K(v_i) \leftarrow \{v_{S_1}, \ldots, v_{S_K}\}$
4: $\quad h_i \leftarrow \{v_i\} \cup \mathcal{N}_K(v_i)$
5: $\quad \mathcal{E}_H \leftarrow \mathcal{E}_H \cup \{h_i\}$
6: **end for**
7: **return** $\mathcal{E}_H$

---

**Algorithm 5** `K-Hop` Lifting

---

**Require:** Adjacency matrix $A \in \{0,1\}^{n \times n}$, hop radius $K$
**Returns:** Hyperedge set $\mathcal{E}_H$
1: $R \leftarrow \sum_{k=1}^{K} A^k$ $\qquad\qquad\qquad\qquad\qquad$ ▷ Reachability matrix up to $K$ hops
2: **for** $i \leftarrow 1$ to $n$ **do**
3: $\quad h_i \leftarrow \{v_j \in V \mid R_{ij} > 0\} \cup \{v_i\}$
4: $\quad \mathcal{E}_H \leftarrow \mathcal{E}_H \cup \{h_i\}$
5: **end for**
6: **return** $\mathcal{E}_H$

---

**Algorithm 6** `Modularity Maximization` Lifting

---

**Require:** Graph $\mathcal{G} = (V, E)$ with adjacency matrix $A \in \mathbb{R}^{n \times n}$, degrees $\{d_i\}_{i=1}^n$, number of communities $N_c$, neighborhood size $K$
**Returns:** Hyperedge set $\mathcal{E}_H$
1: Compute modularity matrix $B \in \mathbb{R}^{n \times n}$ where $B_{ij} = A_{ij} - \frac{d_i d_j}{2|E|}$
2: Compute the top $N_c$ eigenvectors of $B$ and stack them to form $Z \in \mathbb{R}^{n \times N_c}$
3: Cluster rows of $Z$ into $N_c$ communities using K-means
4: **for** each node $v_i \in V$ **do**
5: $\quad$ Pick community $\mathcal{C}$ such that $v_i \in \mathcal{C}$
6: $\quad$ Let $S \leftarrow \{v_j \in \mathcal{C}\}$
7: $\quad$ Compute distances $d(v_i, v_j)$ for all $v_j \in S$
8: $\quad$ Select $K$ nearest neighbors $N_i \subset S$ of $v_i$
9: $\quad$ Define hyperedge $h_{v_i} = N_i \cup \{v_i\}$
10: $\quad \mathcal{E}_H \leftarrow \mathcal{E}_H \cup \{h_{v_i}\}$
11: **end for**
12: **return** $\mathcal{E}_H$

---

**Algorithm 7** `Mapper` Lifting

---

**Require:** Graph $\mathcal{G} = (V, E)$, filter function $g : V \rightarrow \mathbb{R}$, num of intervals $N$, overlap $r \in [0, 1]$
**Returns:** Hyperedge set $\mathcal{E}_H$
1: Compute filter values $g(v_i)$ for all $v_i \in V$
2: Determine range $[a, b] \leftarrow [\min_i g(v_i), \max_i g(v_i)]$
3: Define cover $\mathcal{U} = \{U_1, \ldots, U_N\}$ of $[a, b]$ with overlap ratio $r$
4: **for** each interval $U \in \mathcal{U}$ **do**
5: $\quad$ Define hyperedge $h_U \leftarrow \{v_i \in V \mid g(v_i) \in U\}$
6: $\quad \mathcal{E}_H \leftarrow \mathcal{E}_H \cup \{h_U\}$
7: **end for**
8: **for** each edge $e = (v_i, v_j) \in E$ **do**
9: $\quad \mathcal{E}_H \leftarrow \mathcal{E}_H \cup \{\{v_i, v_j\}\}$ $\qquad\qquad\qquad\qquad$ ▷ Add edges as hyperedges
10: **end for**
11: **return** $\mathcal{E}_H$

---

## B  Complexity analysis

In this section, we analyze in depth the complexity of the discussed liftings. We have also measured the time needed to lift all the datasets, with Table 3 reporting our findings.

**KNN complexity**   The `KNN` lifting requires computing pairwise distances between all node feature vectors to identify the $K$ nearest neighbors for each node. Assuming $n$ nodes with $h$-dimensional features, this involves $O(n^2h)$ operations to compute the full distance matrix, followed by $O(n \log n)$ per node to extract the top-$K$ neighbors, resulting in a total complexity of $O(n^2h + n \log n)$.

**K-Hop complexity**   The `K-Hop` lifting computes for each node the set of nodes reachable within at most $K$ steps using the graph's adjacency matrix. This involves calculating powers of the adjacency matrix up to $A^K$, which can be done efficiently using sparse matrix multiplications when the graph is sparse. Letting $n$ be the number of nodes and $m$ the number of edges, each sparse matrix multiplication takes $O(m)$ time, resulting in an overall complexity of $O(Km)$. Once the $K$-hop neighborhoods are identified, hyperedges are constructed by iterating over the nonzero entries, which also scales with the number of edges traversed. Thus, the total complexity is $O(Km)$.

**EK-Hop complexity**   The `EK-Hop` lifting requires identifying nodes at exact distances 1 through $K$ for each node, which involves computing the powers $A^1, A^2, \ldots, A^K$ of the adjacency matrix. As in `K-Hop`, this has complexity $O(Km)$. However, `EK-Hop` must additionally subtract previously identified neighborhoods at smaller distances to isolate the nodes at exact distance $k$, which adds a small constant overhead per hop.

**FR complexity**   The `FR` lifting first computes the Forman-Ricci curvature for each edge in the graph. For unweighted graphs, the curvature of an edge depends on the degrees of its endpoints and the number of triangles it participates in, which can be computed in $O(m \cdot d_{\max})$ time, where $d_{\max}$ is the maximum node degree. After curvature values are computed, a thresholding operation selects a subset of edges, which is linear in the number of edges. Finally, connected components in the pruned graph are computed in $O(n + m)$ time. The overall complexity is thus dominated by the curvature computation step, resulting in a total complexity of approximately $O(m \cdot d_{\max})$.

**Kernel complexity**   The `Kernel` lifting involves computing a similarity kernel between all pairs of nodes. In the configuration used in this work, it requires computing the graph Laplacian and its spectral decomposition to obtain the heat kernel $\exp(-t\mathcal{L})$, which has a cost of $O(n^3)$ for a full eigendecomposition (where $n$ is the number of nodes). After computing the kernel matrix, identifying the top-$K$ neighbors for each node requires $O(n \log n)$ per node. The overall complexity is $O(n^3 + n \log n)$.

**Mapper complexity**   The `Mapper` lifting first computes a scalar filter function $g : V \to \mathbb{R}$ on all nodes. In this work, $g$ corresponds to the first Laplacian eigenvector. This can be done efficiently using sparse eigensolvers with complexity around $O(m)$ to $O(n^2)$ depending on the graph sparsity and the solver used. Once the scalar values are obtained, the interval $[a, b]$ is partitioned into $N$ overlapping segments, and each node is assigned to the segments (hyperedges) its value falls into, which takes $O(nN)$ time. The overall complexity is thus approximately $O(m + nN)$ assuming sparse graphs and efficient eigenvector computation.

**MM complexity**   The `Modularity Maximization` lifting begins by computing the modularity matrix $B \in \mathbb{R}^{n \times n}$, which has the same sparsity pattern as the adjacency matrix and can be assembled in $O(m)$ time. Spectral clustering on $B$ then requires computing the top-$k$ eigenvectors, where $k$ is the number of desired communities. For sparse graphs, this step typically has a complexity between $O(mk)$ and $O(n^2)$, depending on the eigensolver used. Afterward, K-means clustering over $n$ nodes and $k$-dimensional embeddings incurs $O(nkt)$ time, where $t$ is the number of iterations. Finally, for each node, the $K$ nearest neighbors are computed within its community based on Euclidean distance, which adds up to $O(nKd_c)$, where $d_c$ is the average community size. Overall, the lifting is dominated by the spectral decomposition and clustering, leading to a total complexity of approximately $O(n^2)$ in practice for dense graphs, but lower for sparse inputs.

Table 3: Preprocessing times for the different liftings on the different datasets.

|  | **Cora** | **Citeseer** | **Pubmed** | **Empire** | **Proteins** | **Mutag** | **Amazon** |
|---|---|---|---|---|---|---|---|
| **KNN** | 5.87s | 16.87s | 82.95s | 21.56s | 4.54s | 3.03s | 73.57s |
| **K-Hop** | 3.21s | 3.31s | 13.28s | 12.16s | 4.83s | 3.08s | 27.61s |
| **EK-Hop** | 3.56s | 3.76s | 24.85s | 22.81s | 7.53s | 3.25s | 55.71s |
| **FR** | 2.99s | 3.13s | 4.32s | 3.99s | 3.87s | 3.02s | 5.13s |
| **Kernel** | 6.25s | 7.19s | OOT | OOT | 4.16s | 2.99s | OOT |
| **Mapper** | 3.19s | 3.31s | 9.06s | 151s | 7.17s | 3.18s | 64.45s |
| **MM** | 9.90s | 89.29s | 849s | 1061s | 4.37s | 3.02s | 1234s |

## C HYPERGRAPH MODELS

**AllSetTransformer**  This model is an instance of the AllSet framework Chien et al. (2022), which decouples node and hyperedge updates in message passing. In particular, the hyperedge and node updates can be written respectively as

$$X_h^{t+1} = f_{V \to E_H}(\{X_v^t | v \in h\}, X_h^t),$$
$$X_v^{t+1} = f_{E_H \to V}(\{X_h^{t+1} | v \in h\}, X_v^t),$$

where $X_v^t$ and $X_h^t$ indicate the rows corresponding to node $v$ and hyperedge $h$ of the feature matrix $X^t$. $X_h^0$ and $X_v^0$ correspond to the initial hyperedge and node features. By design, the functions $f_{V \to E_H}$ and $f_{E_H \to V}$ are invariant to permutations of the order of the sets $\{X_v^t | v \in h\}$ and $\{X_h^{t+1} | v \in h\}$. This ensures that the model respects the unordered nature of sets.
The authors propose the AllSetTransformer (AST) model, which incorporates attention to improve the model performance. In particular

$$f_{V \to E_H}(S) = f_{E_H \to V}(S) = \text{LN}(Y + \text{MLP}(Y)),$$

where LN is the layer normalization operation Ba et al. (2016), and Y is obtained from a multihead attention layer Chien et al. (2022). The use of an attention mechanism makes AST effective in capturing complex interactions within hyperedges.

**UniGCNII**  This model is part of the UniGNN framework Huang & Yang (2021), which unifies several GNN architectures under a common design. In particular, this hypergraph model is based on GCN Kipf & Welling (2017), in that it scales how different hyperedges influence a node based on the degree of the nodes in the hyperedge. Inspired by GCNII Chen et al. (2020), this model incorporates initial residual connections and identity mappings to mitigate the over-smoothing problem in deep networks, making it effective for learning in high-order graph structures. The node features at each step are obtained from

$$\hat{X}_v^{t+1} = \frac{1}{\sqrt{d_i}} \sum_{h_i \in \{h | v \in h\}} \frac{1}{\sqrt{d_{h_i}}} X_{h_i}^t,$$
$$X_v^{t+1} = ((1-\beta)I + \beta W)\left((1-\alpha)\hat{X}_v^{t+1} + \alpha X_v^0\right).$$

Here $d_i$ is the degree of node $v_i$, while $d_{h_i}$ is the average degree of the nodes contained in hyperedge $h_i$ while $\alpha$ and $\beta$ are hyperparameters. The matrices $I$ and $W$ indicate, respectively, the identity matrix and a matrix of learnable parameters. In this model, the features of the hyperedges are obtained by averaging the features of the nodes in them.

**ED-HNN**  The Equivariant Diffusion Hypergraph Neural Network (ED-HNN) model Wang et al. (2023) leverages a hypergraph diffusion process that is permutation equivariant over node sets. It applies a principled diffusion operator over the hypergraph structure to propagate information efficiently. At each step, the network calculates the hyperedge and node features as

$$X_h^{t+1} = \sum_{v \in h} \phi(X_v^t),$$
$$X_v^{t+1} = \varphi\big(X_v^t, \sum_{h:v \in h} \rho(X_v^t, X_h^{t+1}), X_v^0, d_v\big),$$

where $\phi$ and $\psi$ are MLPs shared across layers. There are different possibilities for the function $\varphi(\cdot)$, and we use $\varphi(X_v^t, X_v^0) = \text{MLP}((1-\alpha)X_v^t + \alpha X_v^0)$. ED-HNN is specifically designed to respect the symmetry and invariance properties of hypergraphs and has been shown to achieve strong results across various benchmarks.

## D  AVERAGE RANKINGS OF THE LIFTINGS

Table 4: Ranks of the different liftings, for the different models and the overall average rank.

| Model | KNN | K-Hop | EK-Hop | FR | Kernel | Mapper | MM |
|---|---|---|---|---|---|---|---|
| **AST** | 5.14 | 3.42 | 2.57 | 4.29 | 5.57 | **2.00** | 4.14 |
| **UniGCNII** | 4.57 | 2.57 | **2.14** | 2.86 | 5.57 | 4.71 | 4.71 |
| **ED-HNN** | 5.00 | 3.43 | 2.86 | 3.86 | 4.86 | **2.71** | 4.86 |
| **Avg.** | 4.90 | 3.14 | **2.52** | 3.57 | 5.33 | 3.14 | 4.57 |

## E  HYPERPARAMETER SEARCH

Table 5: Best results for KNN lifting.

| Model | Dataset | Best Configuration | Result |
|---|---|---|---|
| AST | **Amazon** | lr=0.01, h=128, K=3 | $45.41 \pm 1.11$ |
| | **Citeseer** | lr=0.001, h=128, K=2 | $68.76 \pm 1.46$ |
| | **Cora** | lr=0.001, h=128, K=4 | $75.16 \pm 1.08$ |
| | **Empire** | lr=0.01, h=128, K=4 | $65.83 \pm 0.41$ |
| | **Mutag** | lr=0.001, h=64, K=3 | $72.77 \pm 5.93$ |
| | **Proteins** | lr=0.01, h=32, K=3 | $76.70 \pm 1.64$ |
| | **Pubmed** | lr=0.01, h=128, K=2 | $87.35 \pm 0.57$ |
| ED-HNN | **Amazon** | lr=0.001, h=128, K=2 | $44.09 \pm 0.76$ |
| | **Citeseer** | lr=0.01, h=128, K=2 | $69.03 \pm 1.01$ |
| | **Cora** | lr=0.001, h=128, K=2 | $72.53 \pm 1.37$ |
| | **Empire** | lr=0.01, h=64, K=2 | $65.66 \pm 0.40$ |
| | **Mutag** | lr=0.001, h=128, K=2 | $80.85 \pm 2.70$ |
| | **Proteins** | lr=0.01, h=128, K=5 | $76.49 \pm 1.43$ |
| | **Pubmed** | lr=0.01, h=32, K=4 | $87.65 \pm 0.42$ |
| UniGCNII | **Amazon** | lr=0.01, h=128, K=2 | $46.88 \pm 0.74$ |
| | **Citeseer** | lr=0.001, h=64, K=2 | $29.60 \pm 0.90$ |
| | **Cora** | lr=0.01, h=128, K=2 | $69.01 \pm 1.48$ |
| | **Empire** | lr=0.01, h=128, K=2 | $65.47 \pm 0.40$ |
| | **Mutag** | lr=0.01, h=32, K=4 | $81.28 \pm 2.49$ |
| | **Proteins** | lr=0.01, h=128, K=2 | $76.13 \pm 1.91$ |
| | **Pubmed** | lr=0.01, h=64, K=3 | $87.44 \pm 0.64$ |

Table 6: Best results for K-Hop lifting.

| Model | Dataset | Best Configuration | Result |
|---|---|---|---|
| AST | **Amazon** | lr=0.001, h=128, K=1 | $48.29 \pm 0.75$ |
| | **Citeseer** | lr=0.001, h=128, K=1 | $74.24 \pm 1.10$ |
| | **Cora** | lr=0.001, h=128, K=1 | $87.36 \pm 0.81$ |
| | **Empire** | lr=0.001, h=128, K=1 | $77.04 \pm 0.85$ |
| | **Mutag** | lr=0.01, h=128, K=2 | $72.34 \pm 7.25$ |
| | **Proteins** | lr=0.01, h=32, K=1 | $76.34 \pm 2.13$ |
| | **Pubmed** | lr=0.01, h=32, K=1 | $89.38 \pm 0.30$ |
| ED-HNN | **Amazon** | lr=0.001, h=128, K=1 | $46.75 \pm 0.87$ |
| | **Citeseer** | lr=0.01, h=64, K=1 | $73.95 \pm 1.26$ |
| | **Cora** | lr=0.001, h=128, K=1 | $86.65 \pm 0.73$ |
| | **Empire** | lr=0.001, h=128, K=1 | $72.81 \pm 0.50$ |
| | **Mutag** | lr=0.001, h=64, K=3 | $71.49 \pm 5.49$ |
| | **Proteins** | lr=0.01, h=32, K=3 | $75.20 \pm 1.25$ |
| | **Pubmed** | lr=0.01, h=32, K=1 | $89.41 \pm 0.29$ |
| UniGCNII | **Amazon** | lr=0.001, h=128, K=1 | $46.09 \pm 0.48$ |
| | **Citeseer** | lr=0.001, h=128, K=1 | $73.45 \pm 1.40$ |
| | **Cora** | lr=0.01, h=64, K=1 | $86.12 \pm 0.76$ |
| | **Empire** | lr=0.01, h=128, K=1 | $74.45 \pm 0.58$ |
| | **Mutag** | lr=0.01, h=128, K=1 | $70.21 \pm 7.00$ |
| | **Proteins** | lr=0.01, h=32, K=1 | $76.49 \pm 1.80$ |
| | **Pubmed** | lr=0.01, h=64, K=1 | $88.50 \pm 0.73$ |

Table 7: Best results for EK-Hop lifting.

| Model | Dataset | Best Configuration | Result |
|---|---|---|---|
| AST | **Amazon** | lr=0.001, h=128, K=1 | $48.38 \pm 0.50$ |
| | **Citeseer** | lr=0.001, h=64, K=1 | $76.30 \pm 4.84$ |
| | **Cora** | lr=0.001, h=32, K=1 | $87.59 \pm 0.80$ |
| | **Empire** | lr=0.001, h=128, K=1 | $77.09 \pm 0.84$ |
| | **Mutag** | lr=0.01, h=128, K=3 | $71.91 \pm 7.30$ |
| | **Proteins** | lr=0.01, h=32, K=1 | $76.34 \pm 2.13$ |
| | **Pubmed** | lr=0.01, h=32, K=1 | $89.45 \pm 0.29$ |
| ED-HNN | **Amazon** | lr=0.001, h=128, K=1 | $46.76 \pm 0.91$ |
| | **Citeseer** | lr=0.01, h=64, K=1 | $73.95 \pm 1.26$ |
| | **Cora** | lr=0.01, h=128, K=1 | $86.88 \pm 1.13$ |
| | **Empire** | lr=0.001, h=128, K=1 | $72.84 \pm 0.53$ |
| | **Mutag** | lr=0.01, h=32, K=2 | $71.06 \pm 7.20$ |
| | **Proteins** | lr=0.01, h=32, K=2 | $75.34 \pm 0.89$ |
| | **Pubmed** | lr=0.01, h=64, K=1 | $89.52 \pm 0.39$ |
| UniGCNII | **Amazon** | lr=0.001, h=128, K=1 | $46.04 \pm 0.60$ |
| | **Citeseer** | lr=0.001, h=128, K=1 | $73.42 \pm 1.42$ |
| | **Cora** | lr=0.01, h=64, K=1 | $86.12 \pm 0.76$ |
| | **Empire** | lr=0.01, h=64, K=1 | $74.29 \pm 0.00$ |
| | **Mutag** | lr=0.01, h=32, K=3 | $82.55 \pm 5.29$ |
| | **Proteins** | lr=0.01, h=32, K=1 | $76.49 \pm 1.80$ |
| | **Pubmed** | lr=0.01, h=64, K=1 | $88.62 \pm 0.43$ |

Table 8: Best results for Kernel lifting. For the feature kernels $\kappa_x$, the possibilities are Identity, Euclidean, and Cosine similarities. For the graph kernels $\kappa_v$, the possible values are Heat, Matérn, and Identity kernels. For the Heat and Matérn kernels, their respective hyperparameters are reported in parentheses.

| Model | Dataset | Best Configuration | Result |
|---|---|---|---|
| AST | Amazon | - | OOT |
| | Citeseer | lr=0.001, h=64, $\kappa_x$=Idn, $\kappa_v$=Heat (T=1) | $72.41 \pm 0.89$ |
| | Cora | lr=0.001, h=128, $\kappa_x$=Idn, $\kappa_v$=Mat ($\nu$=1.5, $\kappa$=0.5) | $86.85 \pm 0.72$ |
| | Empire | - | OOT |
| | Mutag | lr=0.001, h=64, $\kappa_x$=Euc, $\kappa_v$=Heat (T=1) | $72.77 \pm 6.28$ |
| | Proteins | lr=0.001, h=64, $\kappa_x$=Euc, $\kappa_v$=Idn | $75.13 \pm 1.96$ |
| | Pubmed | - | OOT |
| ED-HNN | Amazon | - | OOT |
| | Citeseer | lr=0.001, h=128, $\kappa_x$=Idn, $\kappa_v$=Heat (T=1) | $70.83 \pm 1.37$ |
| | Cora | lr=0.001, h=128, $\kappa_x$=Idn, $\kappa_v$=Heat (T=1) | $85.11 \pm 1.28$ |
| | Empire | - | OOT |
| | Mutag | lr=0.001, h=64, $\kappa_x$=Cos, $\kappa_v$=Idn | $82.55 \pm 3.81$ |
| | Proteins | lr=0.001, h=64, $\kappa_x$=Euc, $\kappa_v$=Mat ($\nu$=1.5, $\kappa$=0.5) | $75.48 \pm 2.39$ |
| | Pubmed | - | OOT |
| UniGCNII | Amazon | - | OOT |
| | Citeseer | lr=0.001, h=128, $\kappa_x$=Idn, $\kappa_v$=Heat (T=1) | $69.34 \pm 1.13$ |
| | Cora | lr=0.001, h=64, $\kappa_x$=Idn, $\kappa_v$=Heat (T=1) | $83.25 \pm 1.48$ |
| | Empire | - | OOT |
| | Mutag | lr=0.001, h=64, $\kappa_x$=Euc, $\kappa_v$=Idn | $80.85 \pm 3.98$ |
| | Proteins | lr=0.001, h=64, $\kappa_x$=Cos, $\kappa_v$=Heat (T=1) | $75.63 \pm 1.54$ |
| | Pubmed | - | OOT |

Table 9: Best results for Forman-Ricci Lifting. The direction (Dir) hyperparameter indicates whether the edges are kept with values below or above the threshold.

| Model | Dataset | Best Configuration | Result |
|---|---|---|---|
| AST | Amazon | lr=0.01, h=128, Thrs=0.95, Dir=Below | $45.55 \pm 0.58$ |
| | Citeseer | lr=0.001, h=128, Thrs=0.5, Dir=Below | $72.10 \pm 0.71$ |
| | Cora | lr=0.001, h=128, Thrs=0.7, Dir=Below | $80.71 \pm 1.15$ |
| | Empire | lr=0.01, h=64, Thrs=0.7, Dir=Below | $69.83 \pm 0.27$ |
| | Mutag | lr=0.01, h=32, Thrs=0.7, Dir=Below | $82.13 \pm 3.71$ |
| | Proteins | lr=0.01, h=32, Thrs=0.9, Dir=Below | $76.34 \pm 1.99$ |
| | Pubmed | lr=0.01, h=32, Thrs=0.9, Dir=Below | $87.56 \pm 0.44$ |
| ED-HNN | Amazon | lr=0.01, h=128, Thrs=0.95, Dir=Below | $45.61 \pm 0.76$ |
| | Citeseer | lr=0.01, h=128, Thrs=0.5, Dir=Below | $70.76 \pm 0.86$ |
| | Cora | lr=0.001, h=128, Thrs=0.7, Dir=Below | $76.43 \pm 0.44$ |
| | Empire | lr=0.01, h=128, Thrs=0.7, Dir=Below | $68.74 \pm 0.52$ |
| | Mutag | lr=0.01, h=64, Thrs=0.95, Dir=Below | $82.55 \pm 2.09$ |
| | Proteins | lr=0.01, h=64, Thrs=0.05, Dir=Above | $76.20 \pm 2.46$ |
| | Pubmed | lr=0.01, h=32, Thrs=0.9, Dir=Below | $87.53 \pm 0.37$ |
| UniGCNII | Amazon | lr=0.01, h=128, Thrs=0.9, Dir=Below | $47.48 \pm 1.29$ |
| | Citeseer | lr=0.001, h=128, Thrs=0.7, Dir=Below | $70.76 \pm 1.82$ |
| | Cora | lr=0.01, h=64, Thrs=0.7, Dir=Below | $78.43 \pm 1.34$ |
| | Empire | lr=0.01, h=64, Thrs=0.7, Dir=Below | $68.07 \pm 0.22$ |
| | Mutag | lr=0.01, h=128, Thrs=0.1, Dir=Below | $84.26 \pm 6.68$ |
| | Proteins | lr=0.01, h=64, Thrs=0.3, Dir=Below | $76.56 \pm 2.32$ |
| | Pubmed | lr=0.01, h=128, Thrs=0.95, Dir=Below | $87.34 \pm 0.27$ |

Table 10: Best results for Mapper lifting. The resolution (Res) parameter indicates in how many intervals the $[a, b]$ interval of values from the filter function is divided into. The Gain parameter indicates the overlap between intervals.

| Model | Dataset | Best Configuration | Result |
|---|---|---|---|
| AST | Amazon | lr=0.001, h=128, Res=20, Gain=0.1, Filter=PCA | $48.46 \pm 0.65$ |
| | Citeseer | lr=0.001, h=128, Res=10, Gain=0.2, Filter=PCA | $74.84 \pm 1.33$ |
| | Cora | lr=0.001, h=128, Res=5, Gain=0.2, Filter=PCA | $86.88 \pm 1.12$ |
| | Empire | lr=0.001, h=128, Res=20, Gain=0.2, Filter=SVD | $80.66 \pm 0.39$ |
| | Mutag | lr=0.01, h=32, Res=10, Gain=0.3, Filter=Sum | $72.77 \pm 5.78$ |
| | Proteins | lr=0.01, h=128, Res=5, Gain=0.2, Filter=Laplacian | $76.63 \pm 1.58$ |
| | Pubmed | lr=0.01, h=32, Res=5, Gain=0.1, Filter=SVD | $89.43 \pm 0.56$ |
| ED-HNN | Amazon | lr=0.01, h=128, Res=20, Gain=0.1, Filter=SVD | $46.07 \pm 0.64$ |
| | Citeseer | lr=0.01, h=128, Res=20, Gain=0.2, Filter=PCA | $74.02 \pm 1.01$ |
| | Cora | lr=0.01, h=128, Res=20, Gain=0.1, Filter=PCA | $84.84 \pm 0.79$ |
| | Empire | lr=0.001, h=128, Res=20, Gain=0.2, Filter=SVD | $74.01 \pm 0.45$ |
| | Mutag | lr=0.001, h=64, Res=10, Gain=0.1, Filter=Sum | $77.87 \pm 4.78$ |
| | Proteins | lr=0.01, h=64, Res=5, Gain=0.3, Filter=PCA | $76.20 \pm 2.07$ |
| | Pubmed | lr=0.001, h=128, Res=20, Gain=0.1, Filter=Sum | $88.05 \pm 0.81$ |
| UniGCNII | Amazon | - | OOT |
| | Citeseer | lr=0.001, h=128, Res=20, Gain=0.1, Filter=PCA | $73.28 \pm 1.39$ |
| | Cora | lr=0.001, h=128, Res=20, Gain=0.1, Filter=PCA | $82.48 \pm 1.45$ |
| | Empire | lr=0.001, h=128, Res=20, Gain=0.1, Filter=PCA | $65.92 \pm 0.25$ |
| | Mutag | lr=0.01, h=32, Res=10, Gain=0.2, Filter=Sum | $81.28 \pm 5.11$ |
| | Proteins | lr=0.01, h=128, Res=5, Gain=0.1, Filter=SVD | $76.34 \pm 1.66$ |
| | Pubmed | lr=0.001, h=128, Res=20, Gain=0.1, Filter=PCA | $43.92 \pm 8.86$ |

Table 11: Best results for Modularity Maximization lifting. The communities (Comm) hyperparameter indicates how many communities to divide the graph into, while K indicates how many neighbors to consider. The Edges parameter indicates whether the initial graph connectivity is added to the final hypergraph as 2-hyperedges or not.

| Model | Dataset | Best Configuration | Result |
|---|---|---|---|
| AST | Amazon | lr=0.01, h=128, Comm=10, K=2, Edges=True | $45.65 \pm 0.55$ |
| | Citeseer | lr=0.001, h=128, Comm=5, K=2, Edges=False | $69.32 \pm 1.10$ |
| | Cora | lr=0.001, h=128, Comm=5, K=5, Edges=True | $78.67 \pm 1.29$ |
| | Empire | lr=0.01, h=64, Comm=2, K=4, Edges=True | $71.46 \pm 0.30$ |
| | Mutag | lr=0.01, h=128, Comm=5, K=3, Edges=True | $73.19 \pm 6.68$ |
| | Proteins | lr=0.01, h=128, Comm=2, K=2, Edges=False | $76.56 \pm 2.30$ |
| | Pubmed | lr=0.01, h=128, Comm=2, K=10, Edges=False | $87.60 \pm 0.40$ |
| ED-HNN | Amazon | lr=0.01, h=128, Comm=5, K=2, Edges=False | $44.48 \pm 0.68$ |
| | Citeseer | lr=0.01, h=128, Comm=5, K=2, Edges=False | $68.84 \pm 1.51$ |
| | Cora | lr=0.001, h=128, Comm=10, K=5, Edges=False | $74.62 \pm 1.26$ |
| | Empire | lr=0.01, h=128, Comm=10, K=2, Edges=True | $67.60 \pm 0.69$ |
| | Mutag | lr=0.01, h=32, Comm=10, K=3, Edges=False | $82.13 \pm 2.17$ |
| | Proteins | lr=0.01, h=64, Comm=2, K=10, Edges=False | $75.77 \pm 1.91$ |
| | Pubmed | lr=0.01, h=32, Comm=2, K=4, Edges=False | $87.75 \pm 0.48$ |
| UniGCNII | Amazon | lr=0.01, h=128, Comm=10, K=2, Edges=False | $46.87 \pm 0.85$ |
| | Citeseer | lr=0.001, h=128, Comm=10, K=2, Edges=True | $35.87 \pm 1.63$ |
| | Cora | lr=0.01, h=64, Comm=10, K=2, Edges=True | $70.84 \pm 1.03$ |
| | Empire | lr=0.01, h=128, Comm=2, K=2, Edges=False | $65.39 \pm 0.39$ |
| | Mutag | lr=0.01, h=32, Comm=2, K=5, Edges=True | $82.13 \pm 4.39$ |
| | Proteins | lr=0.01, h=128, Comm=2, K=2, Edges=False | $76.06 \pm 1.97$ |
| | Pubmed | lr=0.01, h=128, Comm=10, K=2, Edges=False | $87.43 \pm 0.45$ |

