# OpenReview forum: "Lift me up: the impact of liftings on hypergraph neural networks"
_ICLR.cc/2026/Workshop/GRaM — ICLR 2026 Workshop GRaM Poster_

### Official Review · Reviewer_bfcq · 2026-02-13
**A much-needed systematic study of hypergraph lifting heuristics**

**Rating:** 6
**Confidence:** 4

**Review:**

While the hypergraph neural network (HNN) community has focused heavily on architecture, this paper addresses the often-ignored but critical "lifting" process, how we actually construct a hypergraph from a standard graph. The authors provide a systematic evaluation of heuristics like clique-lifting and community detection, and their findings are a significant wake-up call: the choice of lifting often has a larger impact on model performance than the HNN architecture itself. This "service-to-the-field" benchmark provides an essential cheat sheet for practitioners across datasets like Cora and Pubmed. The main limitation, however, is that the study is purely empirical; it identifies that certain liftings outperform others but stops short of providing a theoretical analysis of why (e.g., through hyperedge expansion properties).  Even without a deep theoretical dive, the practical rigor of this benchmark makes it an essential reference that will likely influence how future HNN papers are written.

**Pmlr Suitability:**

No

---

### Official Review · Reviewer_zUrw · 2026-02-21

**Rating:** 7
**Confidence:** 3

**Review:**

The authors provide a systematic evaluation of hypergraph lifting strategies and show that this upstream construction step can be more decisive for downstream performance than the choice of architecture. This is a valuable benchmarking effort and is helpful for researchers and practitioners.

More broadly, the paper’s message resonates with the structure-design perspective seen in sheaf neural networks: 'Sheaf Neural Networks by Hansen et al': in both cases, a large part of the inductive bias is determined before (or jointly with) message passing by how one chooses/learns the underlying geometric/algebraic structure (see 'Neural Sheaf Diffusion: A Topological Perspective on Heterophily and Oversmoothing in GNNs'). As a minor suggestion, it could be useful to add a brief discussion connecting lifting to related structure selection/learning ideas in sheaf-based GNNs, see also: 'Sheaf Neural Networks with Connection Laplacians' and 'Sheaf Attention Networks'. This would help situate the contribution within a broader class of approaches that learn or design the underlying geometric/algebraic structure.

**Pmlr Suitability:**

Yes

---

### Official Review · Reviewer_4Vm1 · 2026-02-24
**Systematic comparison of hypergraph liftings**

**Rating:** 7
**Confidence:** 3

**Review:**

This paper presents a systematic empirical study of hypergraph liftings and their impact on Hypergraph Neural Network (HNN) performance. By evaluating multiple lifting strategies across architectures and datasets, the authors argue that lifting choice often plays a more critical role than architectural design. The paper is well written and well-organized.

Strenghts:
- Provides a systematic empirical analysis on the role of lifting in HNN performance
- Introduces a new lifting (EK-Hop), which demonstrates strong and consistent performance across benchmarks
- Includes a statistical analysis to rigorously compare lifting performance
- Offers a thorough complexity analysis, both theoretical and empirical, highlighting scalability trade-offs.

Weaknesses:
- Finding 1 appears overstated. In 2 out of 6 datasets, GNNs outperform HNNs by a margin of 2-4%, whereas when HNNs perform better, the gain is below 1%, exception made for Mutag.  Moreover, some reported gains over GNNs fall within standard deviation and therefore may not be statistically significant (e.g. on Citeseer).
- The failure on the heterophilic benchmarks is not discussed. It can be that the considered liftings are not suited for heterophilic graphs because they aggregate based on feature/structure similarity, implicitly assuming that such similarity aligns with the label distribution. A discussion of this limitation would help clarify when lifting is beneficial.

Minor comment:
- The reference to "Table 3" at the end of page 3, likely should be "Table 1".

**Pmlr Suitability:**

Yes

---

### Meta-Review · Area_Chair_PuWj · 2026-02-26

**Decision:**

Accept

**Metareview:**

Reviewers agree that this contribution presents a valuable contribution by rigorously comparing hypergraph liftings. This type of paper is needed for the community to make meaningful progress and despite some limitations that I encourage the authors to address, I am happy to recommend acceptance.

**Relevance To Proceedings:**

Yes — suitable for PMLR (long paper)

**Relevance To Workshop:**

Yes — suitable for GRaM

---

### Decision · Program_Chairs · 2026-03-02

Accept (Poster)